# First-Principles Density Functional Theory Calculations of Bilayer Membranes Heterostructures of Ti_3_C_2_T_2_ (MXene)/Graphene and AgNPs

**DOI:** 10.3390/membranes11070543

**Published:** 2021-07-16

**Authors:** Golibjon. R. Berdiyorov, Mohamed E. Madjet, Khaled. A. Mahmoud

**Affiliations:** 1Qatar Environment and Energy Research Institute, Hamad Bin Khalifa University, Doha P.O. Box 34110, Qatar; gberdiyorov@hbku.edu.qa; 2Max Planck Institute for the Physics of Complex Systems Nöthnitzer Str. 38, 01187 Dresden, Germany; mmadjet@pks.mpg.de

**Keywords:** MXene, graphene, DFT, membrane separation, AgNPs

## Abstract

The properties of two-dimensional (2D) layered membrane systems can be medullated by the stacking arrangement and the heterostructure composition of the membrane. This largely affects the performance and stability of such membranes. Here, we have used first-principle density functional theory calculations to conduct a comparative study of two heterostructural bilayer systems of the 2D-MXene (Ti_3_C_2_T_2_, T = F, O, and OH) sheets with graphene and silver nanoparticles (AgNPs). For all considered surface terminations, the binding energy of the MXene/graphene and MXene/AgNPs bilayers increases as compared with graphene/graphene and MXene/MXene bilayer structures. Such strong interlayer interactions are due to profound variations of electrostatic potential across the layers. Larger interlayer binding energies in MXene/graphene systems were obtained even in the presence of water molecules, indicating enhanced stability of such a hybrid system against delamination. We also studied the structural properties of Ti_3_C_2_X_2_ MXene (X = F, O and OH) decorated with silver nanoclusters Ag_n_ (n ≤ 6). We found that regardless of surface functionalization, Ag nanoclusters were strongly adsorbed on the surface of MXene. In addition, Ag nanoparticles enhanced the binding energy between MXene layers. These findings can be useful in enhancing the structural properties of MXene membranes for water purification applications.

## 1. Introduction

MXenes, a growing family of layered low-dimensional transition metal carbides and nitrides, have attracted a lot of attention in the scientific community due to their unique chemical, physical and mechanical properties [1,2]. Electrochemical energy storage applications represent one of the areas where MXenes have already shown great potential due to their impressive capacitive performance [3,4,5,6]. Excellent electrical conductivity and functionalized surface properties make MXene a good candidate for electrochemical sensing applications [7,8,9,10]. MXenes are also shown to be a suitable candidate for nanocomposite fabrication [11,12,13,14]. These composite materials have already shown promising results in terms of both operational stability and functionality (i.e., mechanical strength and flexibility, electrical conductivity) as compared to standalone MXene. For example, the MXene/carbon nanotube hybrid system was shown to be an excellent electrode material for supercapacitors due to its high volumetric capacitances, good operational rate performances, and excellent cycling stability [12].

MXenes have also emerged as high-flux, ultrathin, and size/charge selective nanofiltration membranes for water purification [15,16,17]. As a membrane for water purification applications, this material has already shown good mechanical flexibility, demonstrated thermal and chemical stability, and exhibited large-scale membrane creation and the possibility of surface termination with both hydrophilic and hydrophobic groups, etc. In addition, these low-dimensional materials have profound antibacterial properties which are also beneficial for water treatment [18]. Due to their metallic conductivity, MXenes are also considered to be promising materials for developing highly sensitive electrocatalytic sensors which can detect different chemical contaminants in drinking water, such as bromate [19].

MXene/graphene oxide (GO) hybrid systems have received a lot of interest due to their capacity to deliver enhanced mechanical properties (e.g., tensile strength and elasticity), profound electrical conductivity, and very rich surface chemistries (e.g., the coexistence of hydrophilic and hydrophobic surfaces) [20,21,22]. This hybrid system has the potential to operate as electrode material in electrochemical storage applications due to its fast diffusion and transport of electrolyte ions, high metallic conductivity, and enhanced stability. Such nano-engineered MXene/graphene composites have been used for membrane desalination and water treatment [23,24], as well as for electromagnetic wave shielding applications [25]. In addition to a high surface-to-value ratio and enhanced mechanical strength, MXene/graphene heterostructures provide the coexistence of two types of surfaces which can be beneficial for desalination purposes. Particularly, the hydrophobic surface of graphene may increase the water flux through the system, whereas the hydrophilic surface of MXene will enhance the antifouling properties of the membrane [26]. Due to its chemical stability, graphene layers can also be used to prevent the surface degradation of MXene caused by oxidation. MXene has been used to enforce reduced graphene oxide (rGO) sheets by crosslinking through Ti-O-C covalent bonding to improve the alignment and mechanical properties of graphene platelets. The synergistic interfacial interaction and the resulting high toughness of the hybrid were proven by the in situ Raman spectroscopy and molecular dynamic simulations [27]. Kong et al. have utilized DFT to show a slight distortion in the structure of graphene when forming Ti_3_C_2_O_2_/MXene quantum dots/graphene heterostructures, causing changes to the electronic structure of QDs [28].

Smaller water flux is one of the main issues preventing MXene membranes from being used in large-scale water purification applications. However, the water flux through MXene membranes can be increased significantly by increasing the interlayer spacing (see Ref. [29] for review). This can be achieved by mechanical intercalation of colloidal particles between the MXene layers [30] or by the chemical deposition of metallic nanoparticles during the MXene membrane formation. For example, Pandey et al. decorated the surface of MXene membranes with Ag nanoparticles during the self-reduction of AgNO_3_ on the surface of MXene layers, which acts as a reducing agent) [31]. Depending on the nanoparticle load, the flux in this composite system can be three times larger than the pristine MXene membrane under the same operational conditions. The composite membranes also demonstrated a higher rejection rate for organic molecules/foulants and profound bacterial growth inhibition, indicating their potential for water treatment and biomedical applications. The electrochemical properties of MXene can also be enhanced considerably after decoration with Ag nanoparticles [32]. Rakhi et al. showed the great potential of MXene/Au nanocomposites in developing electrochemical biosensors and in environmental remediation applications [14]. Such MXene/metal nanoparticle hybrid systems are also promising for addressing critical issues in membrane-based separation and biosensor processes [33].

In this work, we have conducted first-principle density functional theory (DFT) calculations to study the structural properties of the MXene/graphene bilayer system with and without the presence of water molecules. Special attention has been given to surface termination of MXene, which is known to be very important in determining both structural and electronic properties of the material [34,35,36,37,38,39]. We considered fluorinated, oxygenated, and hydroxylated surfaces (Ti_3_C_2_F_2_, Ti_3_C_2_O_2_, and Ti_3_C_2_(OH)_2_), which are predominant functional groups of Ti_3_C_2_ MXene [40]. We found that for all considered systems, the binding energy of the graphene/MXene bilayer increases as compared to graphene/graphene and MXene/MXene samples. Such enhanced interlayer interactions originate from the strong variations of electrostatic potential across the layers. Partial charge calculations show the charge transfer from MXene to graphene in the hybrid systems. Stronger interlayer binding energy is also obtained in the presence of water molecules, indicating the enhanced stability of the mixed system against membrane delamination.

Motivated by the recent experiments [31,33], we also studied the structural properties of silver nanoclusters Ag_n_ (n ≤ 6) absorbed on the surface of MXene layers using DFT calculations. We considered smaller nanoclusters (n ≤ 6), which preserve the planar arrangement of silver atoms. We found strong adsorption of Ag_n_ nanoparticles regardless of the surface termination of MXene. The smallest adsorption energies were obtained for fluorinated MXene. In addition, we found that metal clusters enhance the binding energy between MXene layers; this could be useful for increasing the stability of MXene-based membranes against delamination.

## 2. Computational Method

We first optimized the geometry of the freestanding graphene and Ti_3_C_2_ MXene with different surface functionalization, separately using DFT within the generalized gradient approximation of Perdew–Burke–Ernzerhof (PBE) to account for the exchange-correlation energy [41]. The Brillouin zone was sampled using 15 × 15 × 15 k-point sampling [42]. Grimme’s empirical dispersion correction to the PBE was used to account for non-bonding interactions [43]. Double-zeta-polarized basis sets of local numerical orbitals were used, and the electrostatic potentials were calculated on a real-space grid with a mesh cutoff energy of 150 Ry. The convergence criteria for total energy and Hellmann–Feynman forces were 10^−4^ eV and 0.01 eV/Å, respectively. The hexagonal lattice parameter of graphene was 2.478 Å and the MXene samples had the following lattice parameters: 3.126 Å (Ti_3_C_2_F_2_), 3.094 Å (Ti_3_C_2_O_2_) and 3.142 Å (Ti_3_C_2_(OH)_2_). After optimizations, we constructed graphene/graphene, MXene/MXene, and MXene/graphene bilayer structures (see Figure 1a–c). Due to the lattice mismatch, the mean absolute strain in constructing MXene/graphene bilayer systems was less than 1% (see Table 1). Periodic boundary conditions were applied along the planes and a vacuum spacing of 100 Å was always kept across the layers. For the Brillouin zone integration, 7 × 7 × 1 k-points were used.

For MXene/Ag systems, we first optimized isolated silver clusters with up to 6 Ag atoms. We then placed the Ag nanoclusters on the top of 4 × 4 supercell of MXene layers and optimized further using 7 × 7 × 1 k-points. We also constructed MXene bilayers with and without the Ag nanoclusters between them. During the optimization, a vacuum spacing of 100 Å was always kept across the MXene layers. Model system building and calculations were performed using the first-principles computational package Atomistix toolkit [44,45,46]. The analysis in the present work is based on total energy calculations, with a particular focus on the electrostatic contribution. The binding energies between different layers and adsorption energies of the silver clusters are also based on total energy calculations.

## 3. MXene/Graphene Bilayer

First, we tried to minimize the energy of the constructed bilayer systems by varying the interlayer distance *d*. This was performed by explicit calculations for the total energy as a function of bilayer distance. Van der Waals interactions are considered during this energy minimization process. Figure 1d shows the energy of the system as a function of the interlayer distance for graphene/graphene and MXene/graphene systems. Note that the interlayer distance is calculated from the surface atoms of MXene, and the value of the minimal total energy is subtracted from the energy values obtained for a given *d*. Since the considered structures have different interfacial areas (see Table 1), the energies of the system are shown per surface area. The energy of the system shows a minimum at a certain distance, and it increases rapidly as the interplanar distance decreases. The minimum energy distance becomes smaller for MXene/graphene systems compared to that of the graphene/graphene bilayer. At larger distances, we obtained a saturation of the total energy. The maximum saturation value was obtained for the Ti_3_C_2_(OH)_2_/graphene system (open squares in Figure 1d), indicating stronger interlayer interactions as compared to the other hybrid structures. Note that regardless of the surface functionalization, the MXene–graphene distance was larger than the MXene–MXene interlayer distance [1].

Second, we optimized the structures in a more straightforward manner, i.e., by a direct force relaxation, considering van der Waals forces to accurately describe the interlayer interactions. The optimized structures for MXene/graphene bilayers are shown in Figure 2a–c. The optimized interlayer distance values are close to the ones obtained from explicit energy–distance optimization. Figure 2d shows the calculated binding energies of all considered bilayer systems. Among these structures, graphene/graphene bilayer has the smallest binding energy per atom in the system. Among the MXene/MXene structures, Ti_3_C_2_(OH)_2_/Ti_3_C_2_(OH)_2_ system has the largest binding energy. Interestingly, regardless of the surface terminations, MXene/graphene hybrid structures have stronger binding energy as compared to graphene/graphene and MXene/MXene systems, which are highlighted in Figure 2d. The largest binding energy is obtained for hydroxylated MXene.

To find the origin for such strong non-bound interactions between graphene and MXene layers, we have calculated the electrostatic potential in the considered systems. Figure 3 shows the electrostatic potential variations across the layers, averaged over the x–y plane. For the graphene/graphene system (solid-black curve in Figure 3a), we obtained no variations of the electrostatic potential except at the graphene planes. For MXene/MXene bilayers, the smallest variations of the potential were found for the Ti_3_C_2_O_2_/Ti_3_C_2_O_2_ system (dotted green curve in Figure 3a), which explains the smallest binding energy for MXene/MXene systems (see Figure 2d). The amplitude of the electrostatic potential oscillations become more pronounced for the Ti_3_C_2_F_2_/Ti_3_C_2_F_2_ (dashed red curve in Figure 3a) and Ti_3_C_2_(OH)_2_/Ti_3_C_2_(OH)_2_ (dash-dotted blue curve in Figure 3a) systems, which explains the stronger interlayer interaction in these two systems. Interestingly, we obtained much stronger variations of the electrostatic potential across MXene/graphene bilayer structures (Figure 3b). The smallest oscillations were obtained for the fluorinated MXene (dashed red curve in Figure 3b), but they remained larger than the potential oscillations in the Ti_3_C_2_F_2_/Ti_3_C_2_F_2_ system. The largest potential variations were obtained for the hydroxylated MXene, which explains the strongest binding energy of the Ti_3_C_2_(OH)_2_/graphene systems as compared to the other samples (see Figure 2d).

Thirdly, we calculated the partial atomic charge in MXene/graphene bilayer structures using the density-derived electrostatic and chemical (DDEC) charges method [47,48], which is one of the most suitable charge partitioning methods for organic and inorganic hybrid systems [49]. We were mostly interested in the induced charges on carbon atoms on graphene in MXene/graphene structures, keeping in mind the fact of the charge neutrality in the graphene bilayer. The results show that an average charge of 0.0053|e| was induced on the carbon atoms of graphene due to the presence of fluorine groups on MXene. A charge transfer over three times larger was obtained in the case of the Ti_3_C_2_O_2_/graphene system; the average charge of carbon atoms was now 0.0192|e|. Contrary to these two systems, we obtained a charge transfer from the hydroxylated surface of MXene to graphene in the Ti_3_C_2_(OH)_2_/graphene sample. Then, the averaged induced charge on the carbon atoms was −0.0186|e|. Such redistribution of the charges resulted in pronounced electromagnetic potential variations across the MXene/graphene structures (see Figure 3b).

Finally, we studied the binding energies of MXene and graphene layers in the presence of water molecules at the interface between the layers. As a reference, we also studied the structural properties of graphene/graphene bilayer systems. We first conducted geometry optimizations starting from different initial positions and orientations of a single water molecule between the layers. The fully optimized structures are shown in Figure 4a–d. The presence of the water molecule resulted in significant structural changes in the graphene layers of the graphene bilayer (Figure 4a). The water molecules were positioned closer to the MXene layers in MXene/graphene systems and, therefore, the graphene layers were less affected (Figure 4b,c). The orientation of the water molecules also changed depending on the surface functional groups of MXene (see Figure 4d). We have calculated the binding energies of the considered systems as
(1)$$E_{b}=E_{M+g+H_{2}O}-\left(E_{M}+E_{g}+E_{H_{2}O}\right)$$
where *E_M+g+H_2_O_* is the total energy of the total system, E_H_2_O_ is the total energy of isolated water molecules, and *E_M_*/*E_g_* is the total energy of the isolated MXene/graphene bilayer. The calculated binding energies are shown in Figure 4e. Among the considered systems, graphene/graphene bilayer had the smallest binding energy, even in the presence of the water molecule. The interlayer interactions became stronger for MXene/graphene structures. The largest interplanar interactions were obtained for hydroxylated MXene: the binding energy was more than three times larger as compared to the one for the graphene/graphene system. These results indicate the enhanced stability of MXene/graphene layered structures against the delamination that occurred in the presence of water molecules.

## 4. Ag Nanocluster-Decorated MXene

In this section, we study the structural properties of MXene layers decorated with Ag nanoparticles. Experimentally, Ag nanoparticles can be created by self-reduction of AgNO_3_ on the surface of MXene nanosheets (as it acts as a reducing agent) with an average size of 44 nm [31]. Since it is difficult to model such large nanoparticles using the first-principles calculations, we focused on smaller Ag nanoparticles on the surface of MXene. Figure 5a shows the lowest energy configurations of Ag_n_ nanoclusters with n ≤ 6. The considered nanoparticles had planar structures. Interestingly, they preserved their planar geometry even after adsorption on the surface of MXene (Figure 5c,d). For all three surface functional groups, no covalent bonding was obtained between the silver nanoclusters and the surface atoms of MXene. Figure 6 shows the adsorption energies of Ag nanoclusters on the surface of MXene, which were calculated as
(2)$$E_{ads}=E_{M+Ag_{n}}-\left(E_{M}+E_{Ag_{n}}\right)$$
where *E_M_* is the total energy of isolated MXene, *E_Agn_* is the total energy of isolated Ag nanoclusters, and *E_M+Agn_* is the total energy of MXene layer with Ag nanoparticle adsorbed. Although the calculated binding energy values are negative, in what follows we discuss the absolute values of the binding energies. It is seen from this figure that regardless of the surface functional groups, Ag nanoclusters were strongly adsorbed on the surface. More than 0.5 eV adsorption energy per Ag atom was obtained for all considered systems. For all three MXene samples, the adsorption energies increased as the cluster size increased. For all Ag nanoparticles (black columns in Figure 6), the smallest binding energies were obtained for fluorinated MXene. For most of the cluster sizes, the maximum adsorption energies were obtained for the oxygenated sample (red columns in Figure 6). Similar adsorption energies were also obtained for hydroxylated MXene (blue columns in Figure 6).

Next, we studied the effect of Ag_n_ nanoparticles on the binding energy between MXene layers. We first inserted Ag_n_ nanoclusters between two MXene layers of the same surface functional groups for large initial interlayer spacing (d ≥ 6 Å). We then conducted direct force relaxation, considering van der Waals forces to accurately describe the interlayer interactions. The lattice parameters that are perpendicular to the layers were equal to 100 Å during the geometry optimization. Figure 7a–c show the optimized structures of MXene bilayers of different surface functionalization with a Ag_6_ nanocluster between the layers. For the fluorinated sample (Figure 7a), the cluster kept its planar geometry and was located in the middle of the layer with an average distance of 2.63 Å away from the surface fluorine atoms. Similar results were obtained for the hydroxylated MXene, although the distance between surface hydrogen atoms and silver atoms was smaller (1.9 Å). A very different situation was obtained in the case of Ti_3_C_2_O_2_/MXene (see Figure 7b): all the silver atoms covalently bonded with the surface oxygen atoms, and the clustering was not energetically favorable anymore. Similar covalent bonding was also observed for all considered Ag nanoclusters (see the inset in Figure 7d).

Table 2 summarizes our findings of the effect of silver clusters on the structural properties of bilayer MXene samples; the table presents the averaged interlayer spacing for all considered silver clusters. The insertion of the smallest cluster (Ag_2_) increased the interlayer spacing considerably (compare columns two and three). For the fluorinated sample, the interlayer spacing increased only slightly (less than 1%) when the size of the nanocluster was increased. For the Ti_3_C_2_O_2_ sample, the interlayer spacing reduced significantly (~9.5%) when the cluster size was increased. This was due to the formation of covalent bonds between Ag and O atoms. A larger reduction in the interlayer spacing (~10%) was also obtained for Ti_3_C_2_(OH)_2_ MXene when the cluster size was increased (last row in Table 2).

To see how the intercalating Ag nanoclusters affect the interlayer interactions, we have calculated the binding energies of bilayer MXene using the following formula
(3)$$E_{bind}=E_{M+M+Ag_{n}}-\left(E_{M}+E_{M}+E_{Ag_{n}}\right)$$
where *E_M+M+Agn_* is the total energy of the whole system, *E_Agn_* is the total energy of an isolated silver cluster, and *E_M_* is the total energy of the isolated MXene layer. The calculated binding energies are shown in Figure 7d for all considered functional groups. Interestingly, the binding energy slightly increased when Ag nanoclusters were inserted between fluorinated MXene layers (see black columns in Figure 7d). This indicates that the AgNPs have a very small effect on the stability of the material against delamination. However, the nanoparticles enhanced the binding energy for the hydroxylated sample (see blue columns in Figure 7d). The effect became more pronounced for the larger cluster. The largest binding energy among the considered systems is obtained for the Ti_3_C_2_O_2_ MXene. This is due to Ag nanocluster intercalation (red columns in Figure 7d). More than 1 eV gain in energy per silver atom can be obtained. Such strong interlayer interactions are due to the formation of covalent bonds between silver atoms and surface oxygen atoms. Thus, regardless of the surface termination, insertion of silver nanoparticles between the MXene layers increases the interlayer interaction, which should result in the mechanical stability of the system [31] against delamination.

## 5. Conclusions

We have used DFT calculations to study the structural properties of graphene/MXene bilayer structures with different surface terminations of MXene. Regardless of the surface functional groups, stronger interlayer interactions were obtained in these hybrid systems as compared to graphene/graphene and MXene/MXene bilayers. Such enhanced interactions may have originated from the induced charges in the system, which resulted in strong electrostatic potential variations across the system. Larger binding energies were also obtained in the presence of water molecules in the system, indicating the possibility of enhancing the stability of such hybrid systems against delamination. We also studied the structural properties of Ti_3_C_2_X_2_ (X = F, O and OH) MXene decorated with Ag nanoclusters. We found that regardless of surface functionalization, the Ag nanoparticles were strongly adsorbed on the surface of MXene. This is despite non-covalent bonding between the silver and the surface atoms. The AgNPs also increased the binding energy of stacked MXene layers, which could be useful for enhancing the operational stability of MXene membranes for, e.g., water treatment applications.

## Figures and Tables

**Figure 1 membranes-11-00543-f001:**
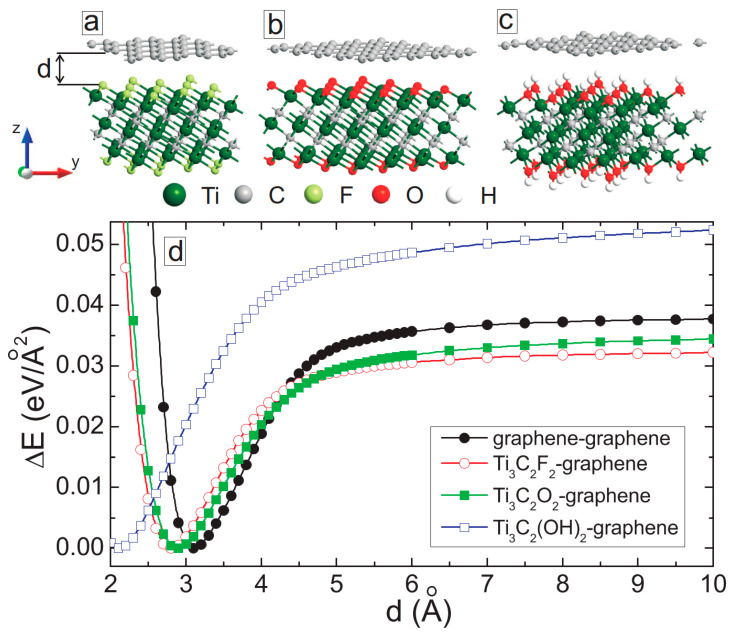
(**a**–**c**) Optimized structures of MXene/graphene bilayers with respect to the interlayer distance *d*. (**d**) The total energy difference of graphene/graphene and MXene/graphene samples as a function of d.

**Figure 2 membranes-11-00543-f002:**
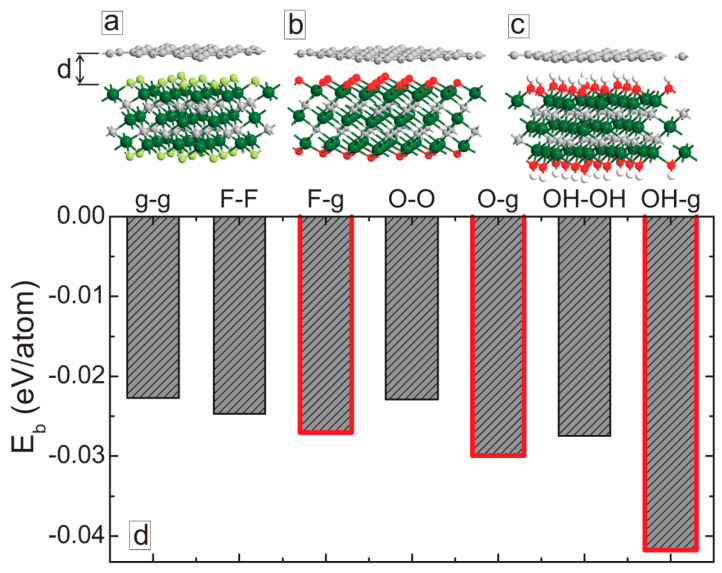
(**a**–**c**) Fully optimized structures of MXene/graphene systems. (**d**) Binding energies (per atom) of graphene/graphene, MXene/MXene, and MXene/graphene bilayer systems. The results for MXene/graphene systems are highlighted by thick red curves.

**Figure 3 membranes-11-00543-f003:**
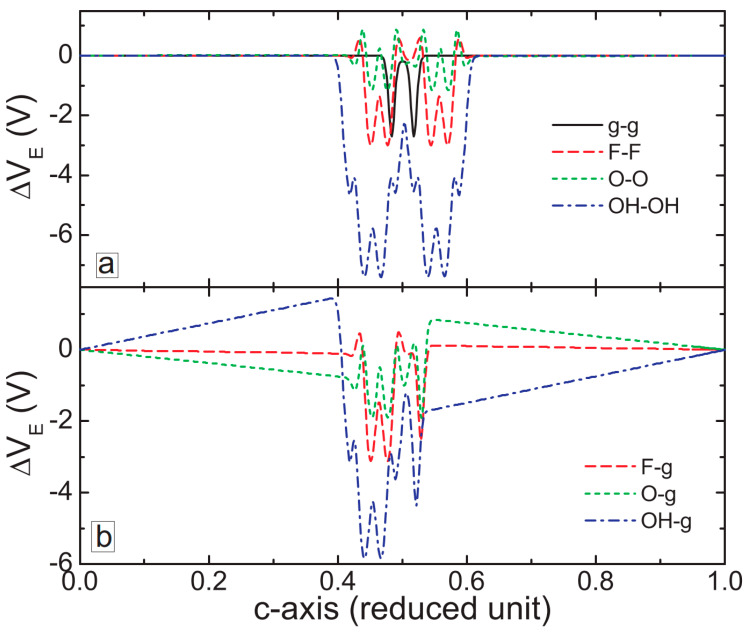
Electrostatic potential variations across the layers of graphene/graphene, MXene/MXene (**a**), and MXene/graphene bilayer samples (**b**).

**Figure 4 membranes-11-00543-f004:**
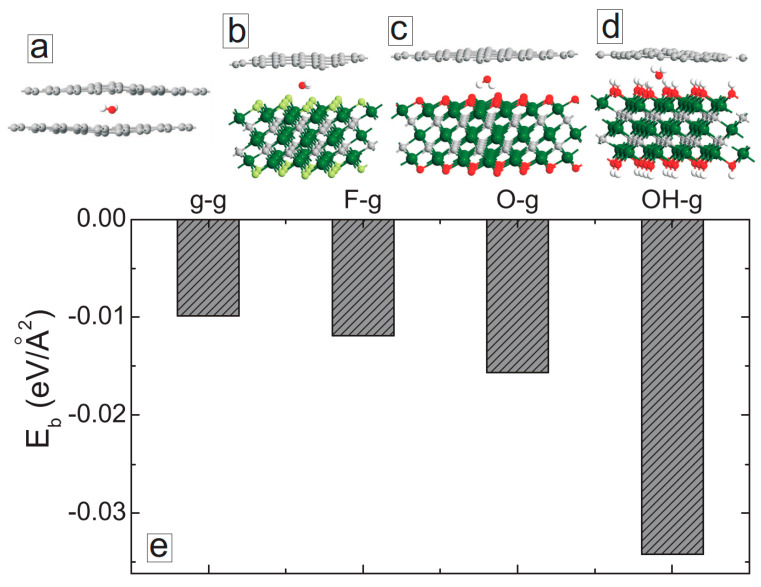
(**a**–**d**) Fully optimized structures of graphene/graphene and MXene/graphene systems in the presence of water molecules. (**e**) Binding energies of graphene/graphene and MXene/graphene systems in the presence of water molecules.

**Figure 5 membranes-11-00543-f005:**
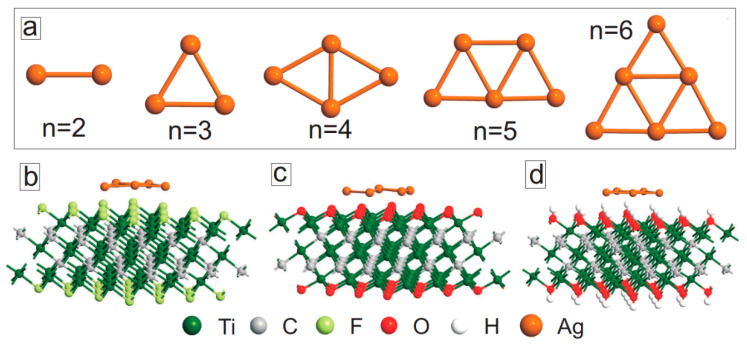
(**a**) Optimized structures of isolated Ag_n_ nanoclusters. (**b**–**d**) Optimized structures of Ag_5_ nanoparticles on Ti_3_C_2_F_2_ (**b**), Ti_3_C_2_O_2_ (**c**) and Ti_3_C_2_(OH)_2_ (**d**).

**Figure 6 membranes-11-00543-f006:**
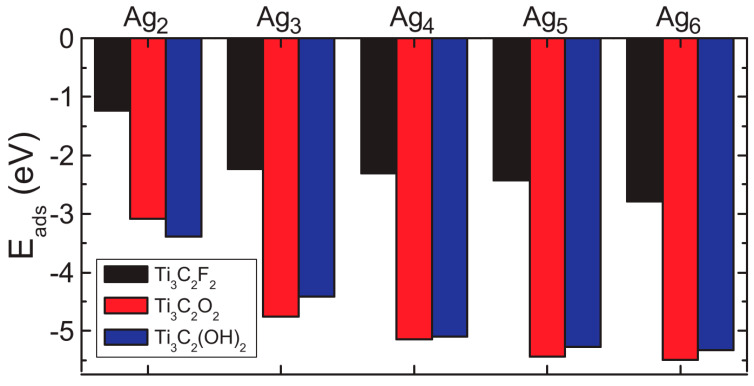
Adsorption energies of Ag_n_ nanoclusters on the surface of Ti_3_C_2_F_2_ (black columns), Ti_3_C_2_O_2_ (red columns), and Ti_3_C_2_(OH)_2_ (blue columns).

**Figure 7 membranes-11-00543-f007:**
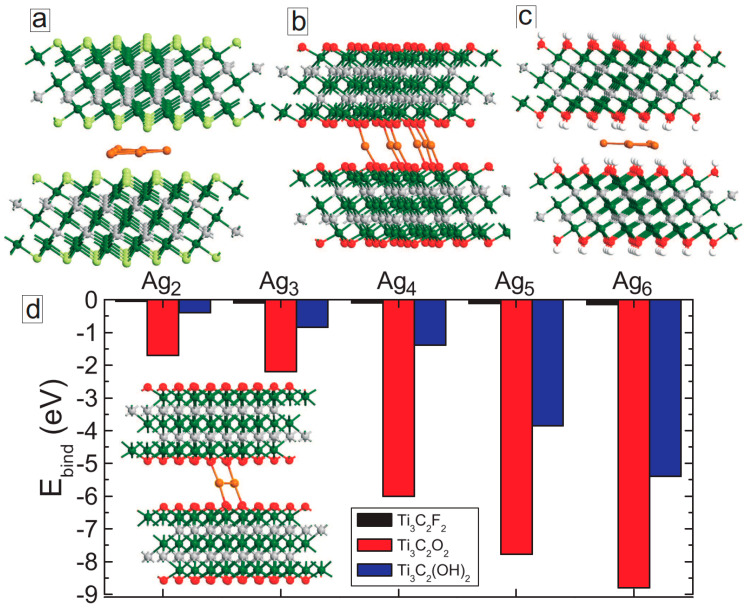
(**a**–**c**) Optimized structures of Ti_3_C_2_F_2_ (**a**), Ti_3_C_2_O_2_ (**b**) and Ti_3_C_2_(OH)_2_ (**c**) bilayer MXene structures with Ag_6_ nanocluster intercalation. (**d**) Binding energies of MXene layers for different Ag_n_ nanoparticles. Inset in (**d**) shows the optimized structure of Ti_3_C_2_O_2_ MXene bilayer with Ag_2_ nanocluster between the layers.

**Table 1 membranes-11-00543-t001:** Structural parameters of the bilayer systems.

Systems	Number of Atoms	Area (Å^2^)	Mean Absolute Strain (%)
graphene/Ti_3_C_2_F_2_	122	101.3	0.09
graphene/Ti_3_C_2_O_2_	162	132.6	0.03
graphene/Ti_3_C_2_(OH)_2_	159	111.4	0.07

**Table 2 membranes-11-00543-t002:** MXene–MXene interlayer spacings (Å) for different Ag clusters.

Systems	No Ag Cluster	Ag_2_	Ag_3_	Ag_4_	Ag_5_	Ag_6_
Ti_3_C_2_F_2_	2.013	4.995	4.981	5.012	4.995	5.011
Ti_3_C_2_O_2_	2.298	4.441	4.017	4.039	4.042	4.043
Ti_3_C_2_(OH)_2_	0.486	4.155	3.845	3.935	3.701	3.704

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
