# Peer review of "First-Principles Density Functional Theory Calculations of Bilayer Membranes Heterostructures of Ti3C2T2 (MXene)/Graphene and AgNPs"

_membranes, 2021, doi:10.3390/membranes11070543_

Round 1
Reviewer 1 Report
This study is an interesting article in which first-principle density functional theory calculations were used to conduct a comparative study of two heterostructural bilayer systems of 14 the 2D-MXene (Ti3C2T2, T= F, O and OH) sheets with graphene and silver nanoparticles (AgNPs). Few minor things should be taken into account:
- The authors should make a table to discuss how the other nanoparticle affects the properties of MXene.
- The study is based on the DFT results. Some other experimental results from the literature should be discussed with DFT results.
Reviewer 2 Report
This work focused on the two hetrostructural bilayer systems of the 2D-MXene (Ti3C2T2, T= F, O and OH) sheets with graphene and silver nanoparticles (AgNPs), the interlayer interactions were investigated by DFT, and the effect of water molecular was discussed. The results were well presented and insterested for the scitific world. However, some modolling conditions need clear modification and explain, I suggested to publicated after minor revision.
- Introduction part need further revision: 1) Line 69-81, the results were showed in Introduction part; 2) the introduction about MXene/Ag was too short; 3) the motivication and the research status were not well presented, especially for the relationship of this two materials to membrane.
- Please explain the selection of k-point sampling.
- Line 118, why atoms less than 6 were investigated?
- Results: different parameters like binding energies, electrostatic potential, total energy were studied, and the results were interesting, it was suggested to explain them combined.
- One single water molecular was used in simulation, is the results credible?
- For the title and the introduction, the author mentioned membrane, but I did not found any information in the simulation process and the results part, it was suggested to enhance the data interpretation from this angle.
